# Effect of Organic Matter Released from Deadwood at Different Decomposition Stages on Physical Properties of Forest Soil

**Wojciech Piaszczyk *** [ID], **Jarosław Lasota and Ewa Błońska**

Department of Ecology and Silviculture, Faculty of Forestry, University of Agriculture in Krakow,
Al. 29-go Listopada 46, 31-425 Kraków, Poland; j.lasota@urk.edu.pl (J.L.); e.blonska@urk.edu.pl (E.B.)
* Correspondence: wojciech.piaszczyk@student.urk.edu.pl

**Abstract:** The wood of dead trees is an essential element of the forest ecosystem, as it affects the characteristics of forest soil properties. The present study aimed to determine the influence of dead alder and aspen wood in various stages of decomposition on the physical properties of forest soil. The study was carried out in the area of the Czarna Rózga reserve in central Poland. Alder and aspen logs in third, fourth, and fifth decay classes were selected for the study. Wood and soil samples under the direct influence of wood and soil samples without the influence of deadwood were collected for laboratory analyses. Physical properties of the soil samples, such as bulk density, moisture, porosity, field capacity, and air capacity were analyzed. Water repellency (WR) was also determined. Our study confirmed that decomposing wood influenced the physical properties of forest soil. Organic matter released from decomposing wood penetrates the soil and alters its physical properties. By releasing organic matter from deadwood, it is possible to stimulate the formation of soil aggregates, improve soil porosity, and significantly increase the number of micropores, which results in the retention of more water in the soil.

**Keywords:** decay class; forest ecosystem; physical properties; soil organic matter

## 1. Introduction

Deadwood is a source of biodiversity and is an important element of the ecosystem that positively affects the physical, chemical, and biochemical properties of soil [1,2]. Wood from dead and fallen trees left in the ecosystem serves as the habitat of fungi, insects, and microorganisms, and it functions as a reservoir of nutrients [3]. Although fungi are the dominant agents of wood decomposition, it has long been known that bacteria also inhabit deadwood [4]. Because of its porous, spongy structure, deadwood constitutes an extremely rich reservoir of water [5], originating both from precipitation and produced directly in the wood through the metabolic processes of bacteria and fungi [6]. Because lying deadwood has direct contact with the soil, soil microorganisms enter the wood and participate in the process of wood decomposition. Consequently, mineral substances accumulated in the wood are transferred to the soil. The main processes that accompany the decomposition of deadwood are respiration, biological transformation, physical decomposition, and leaching [7,8]. Deadwood is a reservoir of macroelements such as nitrogen (N), phosphorus (P), potassium (K), calcium (Ca), and magnesium (Mg), which ultimately end up in the soil. Lasota et al. [9] showed that the wood with the highest decay class releases more ions into the surface layers of the soil than the less decomposed wood. Comparing the ionic composition of water leachate from coniferous and deciduous trees, the latter are characterized by higher cation concentrations than coniferous tree species [9]. The chemical,

biochemical, and physical properties of soil change due to the interaction of the components released from the deadwood [1,10].

To date, more attention has been paid to the impact of deadwood on the chemical and biological properties of soil, while the effect on the physical properties of soil has been less studied. The physical properties of soil play an important role in maintaining favorable conditions for the growth of forest trees. Proper aeration of the soil with favorable moisture conditions is essential for the proper development of the root systems of plants. Maintenance of these conditions is ensured by the appropriate soil structure, which is directly linked to the quantity and quality of soil organic matter (SOM). Organic matter affects all physical characteristics related to soil structure [11]. Celik [12] showed that the conversion of natural pastures to agricultural lands decreased organic matter up to 49% and had crucial effects on the physical properties of soil. A favorable balance between solids and voids in the soil is needed to optimize air exchange, to allow plants to extract water, and to enable the soil to store water in pores. In addition, the organic contents of soil are important in developing the water storage capacity of the surface layers of forest soil [13]. Täumer et al. [14] found a significant correlation between water content, organic matter content, and water repellency (WR) of a soil sample. Soil WR is considered a key soil property that can contribute to the understanding of soil water balance [15,16].

During the decomposition process, deadwood is incorporated into the soil through a process that releases nutrients and organic carbon compounds that significantly influence the soil's C balance and consequently other soil properties [1,8]. The present study mainly aimed to determine the influence of deadwood of various species at various degrees of decomposition on the basic physical properties of soil and the relationship of these characteristics with the chemical properties of soil, especially SOM content.

## 2. Materials and Methods

### 2.1. Sampling Sites and Experimental Design

The investigation was carried out in Czarna Rózga Reserve in Central Poland. The study area is characterized by the following climatic conditions: The average annual rainfall is 649 mm, the average annual temperature is 7.4 °C, and the length of the vegetation season is 200–210 days. Sample plots were located in the area with a predominance of fluvioglacial sand and loam with gleysols, cambisols, and podzols (WRB 2014). The study was carried out in May 2018. Logs of common alder (*Alnus glutinosa*) and common aspen (*Populus tremula*) at the third, fourth, and fifth decay classes (DCs) were selected for the analysis. Deadwood had contact with soil since the beginning of the decomposition process. The DCs of logs were estimated according to the classification of dead trees reported in Maser et al. [17], which was used in previous studies [1,2,9,10] (Table 1). We selected logs with a diameter between 25 and 35 cm to ensure a direct comparison of observations. The soil samples were collected directly under the log, and the soil was sampled from 0 to 10 cm depth. Additional soil samples (background) were taken from the soil at 0–10 cm depth without deadwood influence. In total, 30 samples of wood were collected for the study (2 species × 3 decomposition degrees × 5 replications), and 30 samples of soil under deadwood (2 species × 3 decomposition degrees × 5 replications) and 15 control samples (background) were obtained. Intact soil samples were collected in Kopecky's cylinders. In addition, soil samples with a disturbed structure were collected to determine the texture, pH, and C and N content. Wood was taken from each of the logs analyzed to determine the basic physical and chemical properties (Table 2).

**Table 1.** Decay class characteristics by Maser et al. (1979).

| Degree | Criteria for Evaluation |
|---|---|
| I | Texture intact, circular, natural color of wood, bark intact, branches <3 cm, log leaning on branches |
| II | Texture intact, circular, natural color of wood, bark slightly damaged, no branches <3 cm, log begins to sink |
| III | Texture - larger hard fragments, circular, faded color of wood, fragmented bark, no branches <3 cm, almost entire log on the ground |
| IV | Texture - small pieces, oval shape, faded color of wood, no bark, no branches <3 cm, entire log on the ground |
| V | Texture soft and loose, oval shape, faded color of wood, no bark, no branches <3 cm, entire on the ground |

**Table 2.** Chemical properties of deadwood of different species in different decay classes.

| Species | DC | pH$_{H2O}$ | pH$_{KCl}$ | N | Ct | Lignin |
|---|---|---|---|---|---|---|
| Aspen | III | 4.70 ± 0.62 [a] | 4.18 ± 0.59 [a] | 0.16 ± 0.07 [b] | 45.35 ± 0.24 [a] | 26.99 ± 4.53 [b] |
| | IV | 4.44 ± 0.37 [a] | 3.82 ± 0.40 [a] | 0.66 ± 0.17 [a] | 45.99 ± 1.79 [a] | 37.24 ± 7.74 [a,b] |
| | V | 5.00 ± 0.46 [a] | 4.35 ± 0.58 [a] | 0.88 ± 0.14 [a] | 46.42 ± 0.86 [a] | 37.60 ± 4.75 [a] |
| Alder | III | 4.56 ± 0.99 [a] | 3.86 ± 0.75 [a] | 0.36 ± 0.16 [b] | 47.00 ± 0.43 [a] | 30.48 ± 5.39 [b] |
| | IV | 4.20 ± 0.35 [a] | 3.72 ± 0.28 [a] | 0.49 ± 0.18 [a,b] | 46.32 ± 0.14 [a] | 32.89 ± 1.53 [b] |
| | V | 4.71 ± 0.65 [a] | 4.12 ± 0.60 [a] | 1.10 ± 0.35 [a] | 46.12 ± 0.50 [a] | 40.92 ± 5.53 [a] |

Different lowercase letters in the upper index mean significant differences of parameters between soils under deadwood in different decay classes.

## 2.2. Laboratory Analysis

Soil and deadwood samples obtained in the field were dried and sieved through a 2.0 mm mesh. The basic properties of soil and deadwood samples were determined. The particle size distribution was determined by laser diffraction (Analysette 22, Fritsch, Idar-Oberstein, Germany). By using the potentiometric method, the pH of the samples was analyzed in $H_2O$ and KCl. Carbon (C) and nitrogen (N) contents were measured with an elemental analyzer (LECO CNS TruMac Analyzer; Leco, St. Joseph, MI, USA). Lignin content was spectrophotometrically determined according to the method described in the literature [18,19]. Bulk density, moisture, porosity, field capacity, and air capacity were determined using the Kopecky's cylinder method. The above parameters were calculated using the following formulae:

$$BDa = \frac{c-a}{b} \left[ g/cm^3 \right] \tag{1}$$

where *BDa* is actual bulk density, *a* is the weight of the cylinder [g], *b* is the volume of the cylinder [cm$^3$], and *c* is the weight of the cylinder with soil with natural humidity state,

$$BDd = \frac{e-a}{b} \left[ g/cm^3 \right] \tag{2}$$

where *BDd* is "dry" bulk density and *e* is the weight of the cylinder with dry soil

$$Mw = \frac{c-e}{e-a} * 100 \, [\%] \tag{3}$$

where *Mw* is soil moisture in percent by weight

$$Mv = \frac{c-e}{b} * 100 \, [\%] \tag{4}$$

where *Mv* is soil moisture in volume percent

$$Por = \frac{Dw - BDd}{Dw} * 100 \, [\%] \tag{5}$$

where *Por* is porosity and *Dw* is the density of the solid soil phase

$$CWCw = \frac{d - e}{e - a} * 100 \ [\%]$$ (6)

where *CWCw* is capillary water capacity in percent by weight, and *d* is the weight of cylinder with soil saturated by water.

$$CWCv = \frac{d - e}{b} * 100 \ [\%]$$ (7)

where *CWCv* is capillary water capacity in volume percent.

$$Pa = Por - CWCv \ [\%]$$ (8)

where *Pa* is air capacity.

The density of the solid soil phase (Dw) was established by the pycnometer method. Soil aggregate distribution was determined by dry sieving. By performing dry sieving, four classes of aggregates were separated according to their size (1–2 mm, 2–5 mm, 5–10 mm, and >10 mm). The number of aggregates was converted to 20 g of soil. Water repellency (WR) was determined based on WDPT (water drop penetration time) by using Wessel's test [20]. The test consisted of placing five droplets of distilled water (~0.05 mL) on the soil surface by using a laboratory micropipette in three replicates for each sample. The time (s) needed for the infiltration of water was measured and compared with tabular values according to the classification used by Täumer et al. [14]. Seven classes of repellency were distinguished, ranging from wettable with WDPT less than 5 s to extremely water repellent with WDPT of more than 6 h (Table 3). The experiment was carried out under laboratory conditions at a constant air temperature of 21 ± 1 °C. Water at the same temperature was used to sprinkle the droplets. WDPT analysis was performed in two humidity states: Natural and dry.

**Table 3.** Classes of water drop penetration time (WDPT) used in this study.

| Classes | Class 0 | Class 1 | Class 2 | Class 3 | Class 4 | Class 5 | Class 6 |
|---------|---------|---------|---------|---------|---------|---------|---------|
| | <5 s | 5–60 s | 1–10 min | 10–60 min | 1–3 h | 3–6 h | >6 h |
| WDPT | wettable | slightly water repellent | strongly water repellent | severely water repellent | extremely water repellent | extremely water repellent | extremely water repellent |

*2.3. Statistical Analysis*

ANOVA test was used to evaluate the differences between the mean values of the soil properties for different species and at different decay classes. Moreover, the Pearson correlation coefficients for the soil characteristics were calculated. A general linear model (GLM) was used to investigate the effect of the species, DCs, and carbon content on the dry bulk density. The principal component analysis (PCA) method was used to evaluate the relationships between soil properties. On the basis of Ward's method, agglomeration of the soils samples into groups differing in the bulk density, porosity, moisture, capillary water capacity, aggregate distribution, and WDPT was conducted. Differences with $p < 0.05$ were considered statistically significant. All analyses were performed using Statistica 12 software (StatSoft 2012).

**3. Results**

Under the influence of deadwood, the soil samples showed differences in pH and C and N content (Table 4). The soil samples included in the study had a similar texture. For soil under the influence of aspen wood, no statistically significant differences in pH were observed between the DCs. Statistically significant higher pH was observed in soil under alder in the fourth and fifth DCs. The highest C content was found in the soil under the influence of alder deadwood in the fourth and fifth DCs and

under the influence of aspen deadwood in the fifth DCs (Table 4). Similar relationships were observed for the N content in the studied soil samples (Table 4). Lignin content in wood expressed the degree of decomposition. The findings demonstrate an increase in lignin content in the subsequent decay classes (Table 4).

**Table 4.** pH, carbon and nitrogen content in soil collected under deadwood and from the background (control C).

| Properties | Decay Classes | | | | | | |
|---|---|---|---|---|---|---|---|
| | C | III | IV | V | III | IV | V |
| | Common Aspen | | | | Common Alder | | |
| pH H$_2$O | 4.66 ± 0.76 [a] | 4.49 ± 0.35 [a] | 4.15 ± 0.27 [a] | 4.27 ± 0.32 [a] | 4.89 ± 0.23 [a,c] | 5.43 ± 0.26 [b] | 5.26 ± 0.19 [b,c] |
| pH KCl | 3.94 ± 0.75 [a] | 3.77 ± 0.36 [a] | 3.59 ± 0.24 [a] | 3.29 ± 0.30 [a] | 4.02 ± 0.11 [a] | 4.81 ± 0.16 [b] | 4.62 ± 0.25 [b] |
| N | 0.56 ± 0.23 [a] | 0.61 ± 0.13 [a] | 0.66 ± 0.13 [a,b] | 0.82 ± 0.16 [b] | 0.64 ± 0.23 [a] | 1.12 ± 0.26 [b] | 1.33 ± 0.27 [b] |
| Ct | 8.34 ± 3.52 [a] | 9.99 ± 2.15 [a] | 12.50 ± 2.35 [a] | 16.13 ± 3.15 [b] | 9.18 ± 3.39 [a] | 17.78 ± 4.04 [b] | 20.81 ± 4.27 [b] |
| sand | 69.9 ± 9.0 [a] | 67.0 ± 8.4 [a] | 68.0 ± 7.0 [a] | 65.0 ± 10.4 [a] | 68.0 ± 7.0 [a] | 68.0 ± 8.4 [a] | 67.0 ± 9.6 [a] |
| silt | 27.1 ± 6.0 [a] | 28.5 ± 7.9 [a] | 30.1 ± 5.5 [a] | 28.9 ± 6.0 [a] | 25.8 ± 5.8 [a] | 26.8 ± 7.2 [a] | 27.4 ± 8.2 [a] |
| clay | 2.5 ± 1.8 [a] | 3.3 ± 1.9 [a] | 2.0 ± 1.4 [a] | 3.9 ± 2.2 [a] | 3.7 ± 2.3 [a] | 3.2 ± 2.0 [a] | 3.5 ± 1.5 [a] |

Mean ± SD, Ct—carbon content (%), N—nitrogen content (%), DC—decay classes (III-V), C—control. Different lowercase letters in the upper index mean significant differences of parameters between soils under deadwood in different decay classes.

The present study demonstrated that the bulk density of soil tended to decrease under the influence of wood of both studied species (Table 5). The highest density was observed in the soil from the control plots without wood effect (2.53 g/cm$^3$). The lowest density was observed in the soil affected by the wood in the fifth DC (2.32 g/cm$^3$ for aspen and 1.95 g/cm$^3$ for alder) (Table 5). The percentage moisture content by weight in soil under the influence of aspen wood was the highest in the third DC and amounted to 26.6% by weight, while the lowest percentage moisture content was in the control and amounted to 18.8% by weight. For alder, an increasing tendency in moisture content was observed, the higher decay class, the higher was the moisture content of the soil under the logs (18.8% and 66.2% by weight for the control and soil underwood in the fifth DC). Soil porosity for both species showed an increase, the higher decay class, the more porous was observed in the soil under its influence. The lowest porosity of 61.7% was observed in the soil from the control plots, while the highest porosity was observed in the soil affected by wood in the fifth DC (for aspen and alder 72.9% and 84.4%, respectively) (Table 5). Capillary water capacity by weight percentage for soil under alder and aspen wood was the highest in the fifth DC (267.9% and 89.2% respectively) and the lowest in the control soil (54.7%). The highest capillary water capacity by volume percentage was recorded in soil under the influence of alder wood in the fifth DC (76.7%) and the lowest in the soil under the influence of aspen wood in the fourth DC (48.6%). The aggregation of the soil for both wood species was similar (Table 4). For soils affected by both aspen and alder wood, the most numerous were fine aggregates with a diameter of 1 to 2 mm. The smallest number of aggregates were found for those with diameters above 10 mm. The highest number of small aggregates was found in the soil under the influence of alder well-decomposed wood in the fifth DC, with an average of 2821 pieces in 20 g of soil, while the least number of small aggregates was found in the soil under the influence of aspen wood in the third DC and in the control soil, with an average of 1112 and 1193 pieces in 20 g of soil, respectively. The soils from the control plots were characterized by the shortest water penetration time in two humidity states (natural and dry) (Table 5). In general, the water penetration time was longer in the soil under the influence of more decomposed wood, regardless of soil humidity. This relationship was observed for both wood species (Table 5). A positive correlation between WDPT and organic carbon content in soils under the influence of decomposing wood was noted (Table 6).

**Table 5.** Physical properties of soil collected under deadwood and from the background.

| Properties | Decay Classes | | | | | | |
| | C | III | IV | V | III | IV | V |
| | Common Aspen | | | | Common Alder | | |
|---|---|---|---|---|---|---|---|
| Dw | 2.53 ± 0.05 [a,b] | 2.51 ± 0.21 [a] | 2.47 ± 0.09 [a,b] | 2.32 ± 0.14 [b] | 2.42 ± 0.14 [a] | 2.36 ± 0.14 [a] | 1.95 ± 0.16 [b] |
| BDa | 1.15 ± 0.10 [a] | 1.07 ± 0.11 [a,b] | 0.92 ± 0.12 [b] | 0.76 ± 0.11 [c] | 0.92 ± 0.24 [a,b] | 0.84 ± 0.20 [b] | 0.50 ± 0.10 [c] |
| BDd | 0.97 ± 0.09 [a] | 0.85 ± 0.11 [a,b] | 0.78 ± 0.12 [b] | 0.63 ± 0.10 [c] | 0.75 ± 0.23 [a,b] | 0.63 ± 0.18 [b] | 0.31 ± 0.09 [c] |
| Mw % | 18.8 ± 1.7 [a] | 26.6 ± 4.7 [b] | 19.2 ± 2.9 [a] | 20.7 ± 4.9 [a] | 24.9 ± 7.8 [a,b] | 35.5 ± 8.3 [b] | 66.2 ± 16.4 [c] |
| Mv % | 18.2 ± 1.5 [a] | 22.1 ± 2.3 [b] | 14.6 ± 0.8 [c] | 12.8 ± 2.9 [c] | 17.1 ± 2.6 [a] | 21.1 ± 2.4 [b] | 19.3 ± 1.9 [a,b] |
| Por % | 61.7 ± 3.4 [a] | 66.3 ± 3.3 [a,b] | 68.6 ± 4.0 [b,c] | 72.9 ± 4.2 [c] | 69.5 ± 8.1 [a,b] | 73.7 ± 6.4 [b] | 84.4 ± 3.1 [c] |
| Pa % | 9.1 ± 3.8 [a] | 11.8 ± 4.8 [a] | 20.0 ± 6.8 [b] | 18.6 ± 4.4 [b] | 13.3 ± 9.4 [a] | 13.1 ± 8.5 [a] | 7.8 ± 7.7 [a] |
| CWCw % | 54.7 ± 5.1 [a] | 65.5 ± 10 [a] | 63.6 ± 9.9 [a] | 89.2 ± 22.1 [b] | 82.6 ± 27.4 [a] | 103.4 ± 28.2 [a] | 267.9 ± 86.3 [b] |
| CWCv % | 52.6 ± 0.8 [a,b] | 54.4 ± 2.9 [a] | 48.6 ± 4.6 [b] | 54.3 ± 5.2 [a] | 56.2 ± 5.2 [a] | 60.6 ± 3.1 [a] | 76.7 ± 7.6 [b] |
| A.1–2 mm [pc.] * | 1193.3 ± 304.2 [a,b] | 1112.1 ± 257.1 [a] | 1448.2 ± 364.9 [a,b] | 1677.7 ± 568.9 [b] | 1618.7 ± 491.9 [a] | 1354.6 ± 324.7 [a] | 2821.0 ± 862.3 [b] |
| A.2–5 mm [pc.] * | 350.9 ± 124.4 [a] | 230.2 ± 58.0 [a] | 255.4 ± 75.7 [a] | 237.3 ± 73.8 [a] | 300.6 ± 73.3 [a] | 346.3 ± 114.6 [a] | 337.9 ± 104.4 [a] |
| A.5–10 mm [pc.] * | 11.4 ± 6.5 [a] | 11.9 ± 3.1 [a] | 11.4 ± 6.4 [a] | 14.8 ± 6.7 [a] | 5.7 ± 3.3 [a,b] | 9.4 ± 3.6 [a] | 2.02 ± 4.0 [b] |
| A.> 10 mm [pc.] * | 4.0 ± 4.3 [a] | 10.1 ± 3.6 [b] | 6.6 ± 2.8 [a,b] | 4.5 ± 3.4 [a] | 2.5 ± 1.2 [a,b] | 5.6 ± 3.2 [a] | 1.2 ± 1.9 [b] |
| WDPTd min | 3.33 ± 4. [a] | 10.4 ± 15.1 [a] | 36.3 ± 26.2 [b] | 39.4 ± 16.3 [b] | 18.4 ± 28.3 [a] | 2.9 ± 2.2 [a] | 72.5 ± 38.2 [b] |
| WDPTf min | 0.3 ± 0.3 [a] | 0.6 ± 1.3 [a] | 1.4 ± 1.3 [a] | 23.0 ± 42.2 [a] | 1.4 ± 1.5 [a] | 4.2 ± 10.1 [a,b] | 13.1 ± 8.2 [b] |

Mean ± SD, DC—decay classes (III-V), C—control, Dw—density of the solid phase, WDPTd—water penetration time for dry soils, WDPTf—water penetration time for fresh soils. Different lowercase letters in the upper index mean significant differences of parameters between soils under deadwood in different decay classes. * the number of aggregates has been expressed to 20 g of soil.

**Table 6.** The Pearson correlation coefficients between physical and chemical properties.

|  | pHH$_2$O | pH KCl | N | Ct |
|---|---|---|---|---|
| Dw | −0.272 * | −0.283 * | −0.767 * | −0.744 * |
| BDa | −0.155 | −0.178 | −0.795 * | −0.819 * |
| BDd | −0.227 | −0.257 * | −0.829 * | −0.832 * |
| Ww | 0.414 * | 0.471 * | 0.816 * | 0.725 * |
| Wv | 0.436 * | 0.485 * | 0.077 | −0.067 |
| Por | 0.22 | 0.258 * | 0.829 * | 0.829 * |
| Pa | −0.401 * | −0.381 * | 0.083 | 0.197 |
| CWCw | 0.328 * | 0.364 * | 0.807 * | 0.748 * |
| CWCv | 0.516 * | 0.532 * | 0.657 * | 0.565 * |
| A.1-2 mm | 0.249 * | 0.248 * | 0.488 * | 0.458 * |
| A.2-5 mm | 0.311 * | 0.305 * | 0.096 | 0.007 |
| WDPTf | −0.027 | −0.023 | 0.409 * | 0.463 * |
| WDPTd | −0.083 | -0.029 | 0.196 | 0.248 * |

* $p < 0.05$.

The results of the GLM analysis (Table 7) showed that the bulk density of the dry soil was affected by the decay class ($p = 0.000$) and the carbon content ($p = 0.000$). The soil bulk density is less determined by the wood species (Table 7). The density of the solid phase, bulk density, moisture, porosity, capillary water capacity, the number of small aggregates (from 1 to 2 mm) were correlated with carbon and nitrogen content (Tables 6 and 8). Figure 1 shows the results of the principal component analysis (PCA). Factors 1 and 2 isolated during the analysis explain 59.28% of the variance of the analyzed soil properties. Factor 1 explains 41.61%, and factor 2 explains 17.67% of the variability. The soil under the influence of wood shows a higher content of carbon and nitrogen and more favorable physical properties. The soil under the influence of wood in the third DC was characterized by properties similar to those of the control. The results of cluster analysis show that soil under the influence of more decomposed wood (fourth and fifth DC) differ in physical properties from the control and soil under the influence of less decomposed wood (third DC) (Figure 2).

**Table 7.** Summary of general linear model (GLM) analysis of the effect decay class (DC), species and carbon content (Ct) on bulk density (BDd) (significance effect ($p < 0.05$) are shown in bold).

|  | BDd | |
|---|---|---|
|  | F | *p* Value |
| DC | 14.9516 | **0.00000** |
| Species | 0.8857 | 0.35091 |
| Ct | 68.4818 | **0.00000** |
| DC *species | 9.0507 | **0.00006** |
| DC *Ct | 22.8415 | **0.00000** |
| Species *Ct | 0.0038 | 0.95120 |

**Table 8.** The Pearson correlation coefficients between physical properties.

| | Dw | BDa | BDd | Mw | Mv | Por | Pa | CWCw | CWCv | A.1–2 mm | A.2–5 mm | WDPTf | WDPTd |
|---|---|---|---|---|---|---|---|---|---|---|---|---|---|
| Dw | | | | | | | | | | | | | |
| BDa | 0.851 * | | | | | | | | | | | | |
| BDd | 0.871 * | 0.989 * | | | | | | | | | | | |
| Mw | −0.841 * | −0.737 * | −0.806 * | | | | | | | | | | |
| Mv | 0.024 | 0.260 * | 0.114 | 0.314 * | | | | | | | | | |
| Por | −0.806 * | −0.978 * | −0.989 * | 0.802 * | −0.115 | | | | | | | | |
| Pa | 0.014 | −0.340 * | −0.270 * | −0.151 | −0.523 * | 0.293 * | | | | | | | |
| CWCw | −0.876 * | −0.796 * | −0.829 * | 0.949 * | 0.069 | 0.825 * | −0.165 | | | | | | |
| CWCv | −0.714 * | −0.580 * | −0.646 * | 0.822 * | 0.322 * | 0.637 * | −0.551 * | 0.853 * | | | | | |
| A.1–2 mm | −0.671 * | −0.583 * | −0.603 * | 0.669 * | 0.016 | 0.573 * | −0.075 | 0.671 * | 0.561 * | | | | |
| A.2–5 mm | −0.016 | 0.006 | −0.029 | 0.144 | 0.231 | 0.041 | −0.217 | 0.09 | 0.21 | 0.14 | | | |
| WDPTf | −0.565 * | −0.586 * | −0.578 * | 0.480 * | −0.165 | 0.561 * | 0.062 | 0.545 * | 0.440 * | 0.656 * | 0.017 | | |
| WDPTd | −0.226 | −0.262 * | −0.250 * | 0.17 | −0.126 | 0.246 * | 0.334 * | 0.158 | −0.054 | 0.116 | −0.037 | 0.314 * | |

* $p < 0.05$.

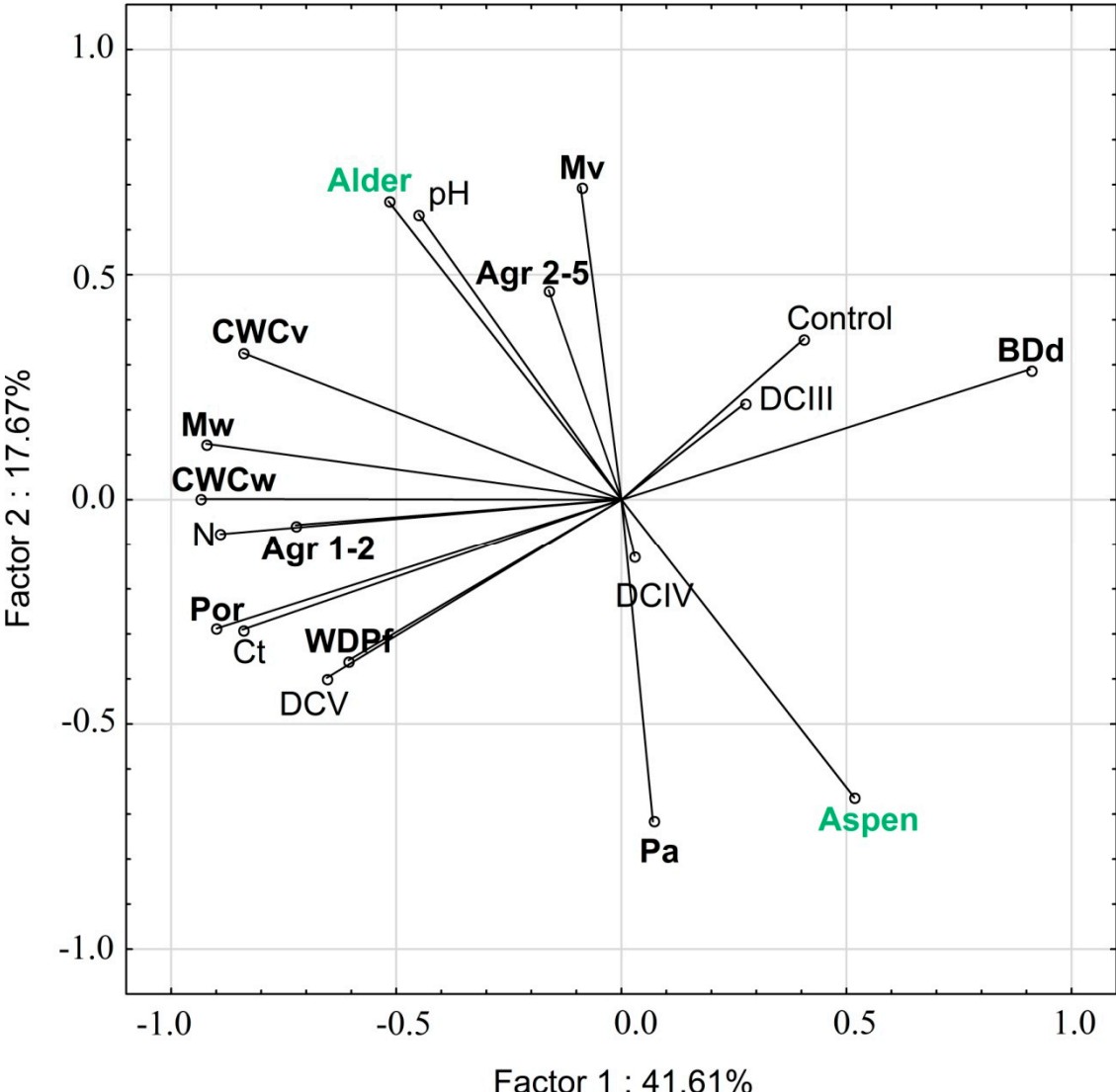

**Figure 1.** Diagram of principal component analysis (PCA) with projection of variables on a plane of the first and second factor for soil (Ct—total carbon content, N—nitrogen content, Por—porosity, Mv—soil moisture in volume percent, Mw—soil moisture in percent by weight, Pa—air capacity, BDd—"dry" bulk density, CWCw—field capacity in percent by weight, CWCv—field capacity in volume percent, WDPTf—water drop penetration time of fresh soils, Agr—aggregate distribution, DC III–V—different decay class).

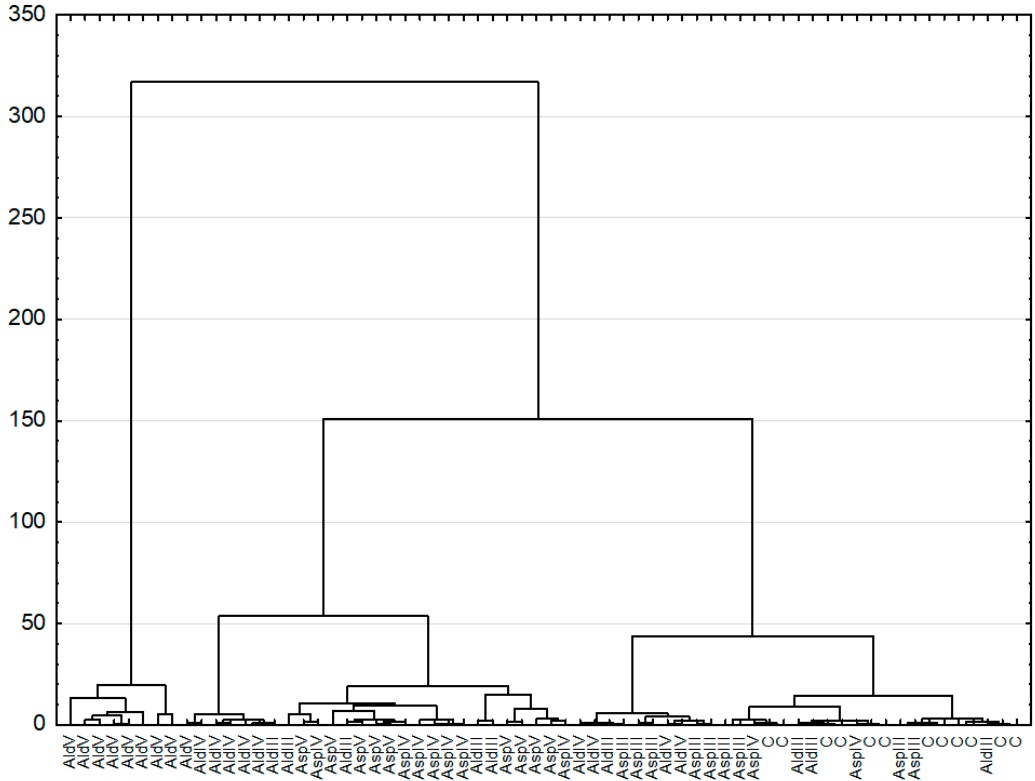

**Figure 2.** Dendrogram with group identified in the cluster analysis (Ald—alder, Asp—aspen, III-V—different decay class, C—control). Bulk density, porosity, moisture, capillary water capacity, aggregate distribution, and WDPT were used for dendrogram.

## 4. Discussion

Our study confirmed the influence of deadwood decomposition on the physical properties of forest soils. Organic matter released from decomposing wood penetrates the soil, thereby altering its physical properties. Leachates from deadwood penetrate the soil via filtering water, whereas fragmented wood penetrates soil through bioturbation as a result of biological activity [21]. Deadwood can change not only carbon storage and nutrient availability in soil but also its water holding capacity [2,22]. The examined physical properties correlated with soil carbon content. In the case of soil density, it was a negative correlation. According to Chen et al. [23], organic matter content has a dominant effect on soil bulk density. In soil science, organic matter concentration is used to predict soil bulk density [24,25]. A positive correlation was observed between moisture, porosity, capillary water capacity or aggregation and organic matter content. Organic remains released from deadwood are delivered to soil and determine soil structure and aggregation. Plant residues stimulate the activity of microorganisms that contribute to the stabilization of soil structure. According to Piaszczyk et al. [10], decaying wood clearly influences the activity of soil enzymes and microbial biomass of C and N in soil under the logs. Enzymatic activity is an indirect measure of the activity of soil microorganisms involved in the decomposition of SOM [26]. The soil structure determines the appropriate air-water properties. The conducted study confirms that the soil under the influence of decomposing deadwood is characterized by more favorable air-water properties. According to Kay and Van den Bygaart [27] and Urbanek and Horn [28], organic matter plays a decisive role in the formation of pores and the stabilization of soil structure. The soil under the influence of decomposing wood differed significantly from the control sample in terms of capillary water capacity, which indicates a significant increase in the number of micropores capable of retaining water. Kirchmann and Gerzabek [29] showed that organic matter supplied to the soil stimulates the formation of micropores by creating an organo-mineral association.

A significant change in physical properties was observed in soil under the influence of strongly decomposed wood (fourth and fifth DC). In soil under the influence of wood in the third DC, no changes with respect to the control were noted. The present study also showed significantly higher carbon content in soil under the influence of strongly decomposed wood (fourth and fifth DC) regardless of the species. Deadwood influences soil properties through its fragmentation and dissolved organic carbon leaching [5,30]. The release of carbon from decomposing wood in successive stages of decay subsequently increases soil water content, porosity, and soil aggregation. A previous study confirmed the intensification of dissolved organic carbon release in the subsequent stages of wood decomposition [5] and an increase in the concentration of cations with the advancement of the deadwood decay class [9]. The cations released from deadwood also affect the physical properties of the soil by shaping the structure. Calcium in the organic matter improves soil aggregation [31]. The wood in the third DC did not influence the physical properties of soils because of a weaker release of substances in the surface soil levels. Wood in its early decay classes has no or little effect on soil properties [32,33].

Decomposition is a long-term process in which the complex structure of wood is reduced to a mineral form through the involvement of microorganisms and invertebrates [34]. The incorporation of decomposing plant residues into the SOM proceeds via their fragmentation by microorganisms-destructors and synthesis of microbial biomass. Slowly and poorly decomposable plant residues represent the main source of particulate organic matter, whereas quickly and highly decomposable plant residues represent the source of microbial biomass [35]. The rate of the wood decomposition process and components released from it depends on several factors such as temperature, humidity, and wood species [9,36]. The studied wood species showed various effects on the properties of the upper soil levels. The quality and quantity of SOM were more favorably influenced by alder wood. Previous studies indicate a beneficial effect of alder wood decomposition on surface soil levels [2,9]. More decomposed alder wood releases more nitrogen compounds to the soil, which is reflected in the biochemical activity of the soils and component circulation. Alder as a sparse species can bind free nitrogen in symbiosis with bacteria in the root, which through decomposing leaves and wood enter the soil [37].

The present study showed differences in water repellency in soil samples under the influence of decomposing wood. An increase in water repellency was observed in soil under the influence of more decomposed wood. Fresh and dry soil samples differed in WDPT values. According to the classification presented by Täumer et al. [14], the penetration time indicates that the fresh soil under influence of deadwood in the fourth and fifth DC is severely water repellent (Class 3). Fresh soil under the influence of wood in the third DC and soil without deadwood influence are slightly and strongly water repellent (class 1 and 2, respectively). The penetration time of dry soil is significantly higher (strongly water repellent—class 2, severely water repellent—class 3, and extremely water repellent—class 4). An increase in WDPT was observed regardless of the species of wood studied. Organic matter released to the soil from decomposed wood affects soil water repellency. Because of wood decomposition, hydrophobic organic compounds are released to the soil. According to Mao et al. [38], the quality of SOM in terms of composition combined with its quality determines the level of soil water repellency. According to Moral Garcia et al. [39], samples with less than 0.06 $g^{-1}$ SOM were slightly to severely water repellent, while those with a higher content of SOM were severely to extremely water repellent.

## 5. Conclusions

Leaving dead trees in the forest ecosystem is justified in terms of its impact on the physical properties of the soil. The present study confirms the positive influence of deadwood in the advanced stage of decomposition on the physical properties of forest soil. To sum up, decaying wood of both investigated species, i.e., alder and aspen, improves air-water properties of soils. Leaving deadwood of these species may be a useful approach for influencing soil C dynamics and soil water capacity. The organic matter released from the deadwood stimulates the formation of soil aggregates, improves

soil porosity, and significantly increases the number of micropores, which results in the retention of more water in the soil. Removing dead trees deprives the soil of recent organic matter and some compounds from decaying wood.

**Author Contributions:** Conceptualization, W.P., J.L. and E.B.; data curation, W.P. and E.B.; formal analysis, W.P., J.L. and E.B.; funding acquisition, E.B.; investigation, W.P., J.L. and E.B.; project administration, J.L.; visualization, W.P.; writing—original draft, W.P. and E.B.; writing—review & editing, W.P., J.L. and E.B. All authors have read and agreed to the published version of the manuscript.

**Funding:** The research was financed by the National Science Centre, Poland (2016/21/D/NZ9/01333).

**Conflicts of Interest:** The authors declare no conflict of interest.

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
