# Peer review of "Effect of Organic Matter Released from Deadwood at Different Decomposition Stages on Physical Properties of Forest Soil"

_forests, doi:10.3390/f11010024_

Round 1

Reviewer 1 Report

General comments

This manuscript describes an observational measurements of physical properties of forest soil with and without dead wood of different species and decay stages. The determination of the effect of dead wood on the basic properties of soil is important to evaluate how to contribute of dead wood to forest ecosystem services like water holding capacity and nutrient supply made by dead wood. The study appears to be well done. However, the ms has many mistakes and insufficient explanations. For example, when you explained about moisture content (L159) you need to clarify volume, weight or both water content is focused. These lacks make reviewer to waste time to review. Your article should be checked carefully by authors and get some English proofreading.

Specific comments

L26: Specify the meaning of “a model for observing natural processes in the environment”.

L74: The decay class of deadwood is important information to understand your study. Adding more explanations about how to estimate decay classes in addition to using reference would be helpful.

L86: In Table 1, you need to use the same order of the tree species to Table 3 and 4. And what is the meaning of “t” in “Ct”? The lowercase alphabet “a” in the lines of “Alder 5” and rows of “Ct” should be smaller.

L118: “Dw” should be added after “The density of the solid phase”.

L141: The number of table referred should be added at the end of the sentence.

L145: In Table 3, there is the significant difference in Ct only in decay class 5 of aspen. The explanation should be changed to be adequate.

L155: The phrase of “mean value” in the parentheses is unnecessary.

L158: The value of moisture content would be better to show in 3 digits.

L159-161: Add “by weight” after moisture content.

L161: Add space before “18.84”

L163: Add “observed in” before the soil under…

L166: Why did you mention the values of capillary water capacity aspen only? Alder has the value more than 200%. Explanation about this should be needed.

L176: Aspen decay class 3 shows the smallest value of 1112, so the explanation is not correct.

L177: Mention more specific about “moisture options”.

L180: WDPT has two values of d and f. Which one you explain should be clarified.

L183: In Table 4, add the explanation of Dw, WDPTd and WDPTf.

L189: In Table 5, change “,” to “.”.

L191: You should indicate “bulk density of the dry soil”.

L195: You should change the order to “Tables 5 and 7”.

L195: Figure 1 is the result of the CCA analysis and you showed only the range of variation explained by the 1st and 2nd axis. I think Figure 1 would be necessary because it doesn’t include important information.

L205: You sometimes use decomposition stage instead of decay class. You should unify the expression to decay class.

L208: You just used Table 7 to explain the positive relationships between the number of the particle size in 1-2mm to other factors. So the table is recommended to show as supplemental information. The second CWCw should be changed to CWCv.

L212: You need to add the explanation of Ct and N in the figure caption.

L216: “Decomposition rate” should be changed to “decay class”.

L219: Change to Figure 2. You should add how to obtain the results of Figure 2 in the method section.

Reviewer 2 Report

This paper describes the results about the influence of various deadwood species at different degrees of decomposition on the basic physical and chemical properties of soil. In my opinion, this manuscript contains original data which is also properly compared with the available literature and the subject seems adequate for Forest Science Journal. The title clearly reflects the content of the manuscript and its abstract. Keywords are sufficiently informative. All results supported by a proper statistical analysis and clearly represented. Nevertheless, I have a few comments and constructive suggestions.

1. It would be useful to describe briefly how the wood samples of the 3rd, 4th, and 5th decay classes differ in morphological features.

2. When were soil sampled? How long have the logs been in contact with the soil? Have the changes in soil properties occurred during the third, fourth and fifth stages of decomposition of wood, respectively or after an application the logs of different stages of decomposition? These questions are important for the understanding of this experiment.

3. The decomposition of wood residues is usually accompanied by an increase in nitrogen content and a decrease in organic carbon. Table 1 shows that the content of Corg remains unchanged, although the amount of lignin (this is correct and OK) increases significantly. Corg data are incomprehensible and may contain a mistake.

4. The percentage of mass aggregates from soil mass but not the number of aggregates of different size classes should be given.

5. Which parameters were expressed the degree of decomposition of the logs?

6. What organic matter could enter the soil from decomposable wood? Soluble organic matter or as particulate organic matter?

7. In the discussion, it would be important to note that changes in the physical properties of the soil can also be caused by the positive effect of organic substances released during the decomposition of wood residues on the soil microbial biomass. Therefore, it would be advisable to add at least two references to the discussion. For example, a) Berg B., McClaugherty C. (2014). Decomposition as a Process: Some Main Features. In: Plant Litter. Springer, Berlin, Heidelberg. DOI https://doi.org/10.1007/978-3-642-38821-7_2   

b) Semenov et al. (2019). Plant Residues Decomposition and Formation of Active Organic Matter in the Soil of the Incubation Experiments. Eurasian Soil Science, 2019, Vol. 52, No. 10, pp. 1183–1194. DOI: 10.1134/S1064229319100119

Best regards.

Round 2

Reviewer 1 Report

No comments.